# A ROR1 Small Molecule Inhibitor (KAN0441571C) Induced Significant Apoptosis of Mantle Cell Lymphoma (MCL) Cells

**DOI:** 10.3390/pharmaceutics14102238

**Published:** 2022-10-20

**Authors:** Amineh Ghaderi, Wen Zhong, Mohammad Ali Okhovat, Johanna Aschan, Ann Svensson, Birgitta Sander, Johan Schultz, Thomas Olin, Anders Österborg, Mohammad Hojjat-Farsangi, Håkan Mellstedt

**Affiliations:** 1Department of Oncology-Pathology, BioClinicum, Karolinska University Hospital Solna and Karolinska Institutet, 171 64 Stockholm, Sweden; 2Department of Laboratory Medicine, Division of Pathology, Karolinska Institutet, 171 77 Stockholm, Sweden; 3Kancera AB, Nanna Svartz Väg 4, 171 65 Solna, Sweden; 4Department of Hematology, Karolinska University Hospital Solna, 171 77 Stockholm, Sweden

**Keywords:** MCL, ROR1, small molecules, apoptosis, targeted therapy

## Abstract

The receptor tyrosine kinase orphan receptor 1 (ROR1) is absent in most normal adult tissues but overexpressed in various malignancies and is of importance for tumor cell survival, proliferation, and metastasis. In this study, we evaluated the apoptotic effects of a novel small molecule inhibitor of ROR1 (KAN0441571C) as well as venetoclax (BCL-2 inhibitor), bendamustine, idelalisib (PI3Kδ inhibitor), everolimus (mTOR inhibitor), and ibrutinib (BTK inhibitor) alone or in combination in human MCL primary cells and cell lines. ROR1 expression was evaluated by flow cytometry and Western blot (WB). Cytotoxicity was analyzed by MTT and apoptosis by Annexin V/PI staining as well as signaling and apoptotic proteins (WB). ROR1 was expressed both in patient-derived MCL cells and human MCL cell lines. KAN0441571C alone induced significant time- and dose-dependent apoptosis of MCL cells. Apoptosis was accompanied by decreased expression of MCL-1 and BCL-2 and cleavage of PARP and caspase 3. ROR1 was dephosphorylated as well as ROR1-associated signaling pathway molecules, including the non-canonical WNT signaling pathway (PI3Kδ/AKT/mTOR). The combination of KAN0441571C and ibrutinib, venetoclax, idelalisib, everolimus, or bendamustine had a synergistic apoptotic effect and significantly prevented phosphorylation of ROR1-associated signaling molecules as compared to KAN0441571C alone. Our results suggest that targeting ROR1 by a small molecule inhibitor, KAN0441571C, should be further evaluated particularly in combination with other targeting drugs as a new therapeutic approach for MCL.

## 1. Introduction

Receptor tyrosine kinases (RTKs) are therapeutic targets in cancer. The RTK ROR1 plays a crucial role in the development of different tissues and organs during embryogenesis [1]. ROR1 is essentially not expressed in normal adult postpartum tissues but overexpressed in several malignancies [2]. Mutations, translocations, deletions, and overexpression of RTKs have been identified. Overexpression of ROR1 in patients was first described in chronic lymphocytic leukemia (CLL) by applying gene expression profiling [3]. ROR1 has also been shown to be overexpressed in several other hematological malignancies, including mantle cell lymphoma (MCL), as well as in solid tumors [4,5,6,7,8,9,10,11,12]. Small molecules and monoclonal antibodies have been used to target dysregulated RTKs in malignancies in a therapeutic approach [13,14].

In neoplastic cells, ROR1 is of importance for survival, proliferation, migration, and metastasis [15]. ROR1 mediates chemotactic and proliferative signals induced by the binding of WNT5a to ROR1 [16], resulting in the activation of guanine exchange factors and the phosphoinositol-3 kinase (PI3K)/JNK signaling pathway [17]. Activation of the canonical and non-canonical WNT signaling pathways correlated with activated ROR1 [18]. Moreover, high expression of ROR1 was associated with disease progression and short survival in several malignancies [15,19,20].

Hematological malignancies account for around 6.5% of cancers around the world [21]. Various lymphomas, including MCL, mostly spread to the bone marrow, and different RTKs, such as focal adhesion kinase (FAK), are involved in tumor–stromal cells interaction (in the tumor microenvironment) to enhance tumor cell survival and drug resistance [22]. MCL is a rare but aggressive B-cell lymphoma characterized by the chromosomal translocation (11;14) (q13; q32) and constitutive overexpression of cyclin D1 [23,24] contributing to the uncontrolled growth of MCL cells [25]. Despite the introduction of novel drugs such as BTK inhibitors and new therapeutic principles, including CAR-T, drug resistance and relapses occur, and MCL remains incurable [26]. Thus, there is a need for novel therapeutic approaches with other mechanisms of action (MOAs) than those in clinical use. ROR1 is an important molecule for the malignant phenotype, e.g., tumor cell proliferation, survival, epithelial–mesenchymal transition (EMT), migration, and metastasis [15,26,27], and been suggested to be an interesting target molecule for precision cancer medicine (PCM).

In the present study, the expression of ROR1 in MCL cells was characterized and the apoptotic effects of a novel ROR1 inhibitor, KAN0441571C, were evaluated in in vitro and ex vivo preclinical models as well as in combination with therapeutics with other MOAs of clinical relevance (ibrutinib, idelalisib, bendamustine, everolimus, and venetoclax) that are currently used for the treatment of patients with MCL [28] or have shown promising results in clinical trials. The results indicate that a small molecule ROR1 inhibitor might be a promising new drug candidate that warrants further evaluation in preclinical models and clinical trials in difficult-to-treat MCL patients.

## 2. Materials and Methods

### 2.1. Patients

Patients were diagnosed and treated according to the Swedish MCL guidelines at the Department of Hematology, Karolinska University Hospital, Stockholm, Sweden. The study was authorized by the National Ethics Authority (www.etikprovningsmyndigheten.se (accessed on 20 December 2016)). A written consent form was collected from patients before blood sampling. Eleven patients with MCL were included, consisting of two females and nine males with a median age of 70 years (range 62–84). Five had early-stage untreated MCL and six had active MCL, and all but two were treatment-naïve. The selection was based on the availability of stored blood MCL cells (liquid nitrogen).

### 2.2. Cell Lines

Five human MCL cell lines, Granta-519 (ACC 342), Mino (ACC 687), JeKo1 (ACC 553), JVM-2 (ACC 12), and Z-138 (CRL-3001), obtained from the American Type Culture Collection (ATCC), were used. All cell lines were tested for mycoplasma contamination. The cell lines were cultured in RPMI-1640 medium (Sigma, St. Louis, MO, USA) with 10% fetal bovine serum (FBS) (Gibco, Life Technologies, Karlsruhe, Germany), 100 ug/mL penicillin/streptomycin (Biochrom KG, Berlin, Germany), and 2% glutamine (Biochrom KG) in humidified air at 37 °C with 5% CO_2_.

### 2.3. ROR1 Small Molecule Tyrosine Kinase Inhibitor (KAN0441571C)

The development of the first small molecule ROR1 tyrosine kinase inhibitor (KAN0439834) was reported in 2018 [29]. The second generation of a ROR1 small molecule inhibitor, KAN0441571C, with a higher cytotoxic potency against tumor cells in vitro, as well as longer halftime (T1/2) in a mouse model (11 h) compared to KAN0439834 (2 h), was recently described [30,31] and was applied in the present study.

### 2.4. Cell Surface Marker Analysis (Flowcytometry)

Analysis of surface ROR1 expression was performed using an anti-ROR1 monoclonal antibody (Miltenyi Biotec, Bergisch Gladbach, Germany) [27]. Cells were analyzed by a FACS Canto II flow cytometer (BD Biosciences, San Jose, CA, USA). The FlowJo software program (Tree Star Inc., Ashland, OR, USA) was applied for analysis of cells.

### 2.5. Annexin V/PI Apoptosis Assay

Human MCL cell lines were plated in 24-well plates at a concentration of 5 × 10^4^ cells per well in RPMI-1640 medium containing 10% FBS and incubated with KAN0441571C or venetoclax (BCL-2 inhibitor), ibrutinib (BTK inhibitor), idelalisib (PI3k inhibitor), everolimus (mTOR inhibitor), and bendamustine (an alkylating agent) as single drugs or in combination with KAN0441571C. After 24 h of incubation, the cells were centrifuged and resuspended in 100 μL of Annexin V binding buffer (BD Biosciences) containing 1 μL fluorescein isothiocyanate (FITC)-conjugated Annexin V and 1 μL propidium iodide (PI) (BD Biosciences) and incubated for 15 min at RT.

Frozen primary MCL cells were thawed at 37 °C for 1–2 min and resuspended in 10 mL of RPMI-1640 medium with 10% FBS. Cell viability was analyzed by EVE™ Automatic Cell Counter NanoEnTek (VWR, Atlanta, GA, USA) according to the manufacturer’s description. Cells were cultured in RPMI-1640 medium supplemented with 10% FBS and 20% conditioned media (CM). CM was harvested from 70% confluent HS-5 cells (CRL-11882, ATCC) grown in RPMI-1640 medium with 10% FBS for 72 h. The supernatant was collected, filtered, and used fresh. Then, 2 × 10^5^ MCL cells were seeded in 96-well plates and incubated at 37 °C in 5% CO_2_ for 24 h with and without drugs. Cells were stained with Annexin V/PI gated for CD45^+^/ROR1^+^/CD3^−^ cells, as exemplified in Appendix A. Cells were analyzed with NovoCyte Advanteon flow cytometry (Agilent, Santa Clara, CA, USA), and data were collected with the NovoExpress^®^ 1.4.1 software program (Agilent, Santa Clara, CA, USA).

### 2.6. MTT Cytotoxicity (or % Inhibition of Cell Proliferation or Cell Growth) Assay

MCL cell lines (10^4^ cells/well) were incubated in quadruplicates in 200 µL of RPMI-1640 containing 10% FBS in 96-well plates for 24–72 h at 37 °C with 5% CO_2_. Then, 20 µL MTT solution (5 mg/mL) was added to each well and incubated for another 4 h at 37 °C. Stopping solution (solubilizing reagent) (10% sodium dodecyl sulphate (SDS) in 0.01M HCL) was added to a final volume of 100 µL/well. After overnight incubation at 37 °C, the plates were read in a microplate reader at 570 nm.

### 2.7. Drug Combination Analyses

Zero Interaction Potency (ZIP) is a reference model [32] for evaluating drug combinations by comparing the deviation in potency of the dose–response curves between the individual agents and the combinations. A ZIP delta score (Δ) for each pair of drugs was used to quantify the deviation by taking the average of all delta scores over their dose combinations. Summary synergy scores are defined as the average excess response of drug interactions, e.g., a score of 20 corresponds to 20% of response above the expectation, i.e., synergism. A score close to 0 gives low confidence for synergism or antagonism. A score less than −10 indicates that the relationship between two drugs is possibly antagonistic; from −10 to 10, the relationship between two drugs might be additive; and if the score is >10, the interaction between two drugs is likely to be synergistic [32].

### 2.8. Western Blot (WB) Analysis

Western blot was carried out as previously described [26]. MCL cells were lysed on ice for 30 min in lysis buffer containing 1% Triton X-100, 150 mM NaCl, 5 mM EDTA, and 50 mM Tris-HCl with 1% protease inhibitor cocktail (Sigma-Aldrich, St. Louis, MO, USA) and phosphatase inhibitors (Roche Ltd., Basel, Switzerland) and centrifuged at 14,000 rpm. Protein concentration was measured by Pierce™ BCA Protein Assay Kit (Thermo Fisher Scientific, Waltham, MA, USA). Then, 10–20 µg protein was loaded into each well for WB analysis. The following antibodies were used for the detection of total (t) and phosphorylated (p) proteins: ROR1 (R & D Systems, Minneapolis, MN, USA), own produced polyclonal rabbit anti-p-ROR1 (Tyr 641 and 646, Ser 652) [26], AKT and p-AKT (Ser 473), mammalian target of rapamycin (mTOR) and p-mTOR (Ser 2448), phosphoinositide 3-kinase (PI3K) p110δ, cleaved poly ADP ribose polymerase (PARP), myeloid cell leukemia-1 B-cell lymphoma (BCL)-2, cleaved caspase 3 (Cell Signaling Technology, Danvers, MA, USA), and p-PI3K p110δ (Tyr 585) (Santa Cruz Biotechnology, Santa Cruz, CA, USA). β-actin (Sigma-Aldrich, St. Louis, MO, USA) was used as a loading control. Densitometry measurements of proteins were calculated by Image J1.44p software (National Institute of Health, Bethesda, MD, USA).

### 2.9. Statistical Analysis

Statistical analysis was performed using GraphPad Prism software (GraphPad Software, Inc., La Jolla, CA, USA). EC_50_ values were calculated from the dose–response curve by non-linear curve fitting (HillSlope). The Mann–Whitney U test was applied for comparison of non-Gaussian distributed data. *p*-values ˂ 0.05 were considered significant. Asterisks represent *p*-values of * 0.01 to 0.05, ** 0.001 to < 0.01, *** < 0.001, and **** < 0.0001.

## 3. Results

### 3.1. Surface ROR1 Was Heterogeneously Expressed on MCL Cell Lines and Primary MCL Cells

The frequency of surface ROR1-positive cells of JeKo-1, Mino, Z138, and Granta-519 cell lines was 100%, 99%, 71%, and 21%, respectively, while JVM-2 did not express surface ROR1 (Appendix A). However, in WB, JVM-2 showed intense expression of ROR1 (Appendix A). This indicates that JVM-2 expressed ROR1 as a splice variant lacking the extracellular part. Surface ROR1 was also analyzed in MCL cells from peripheral blood of leukemia patients (n = 11). MCL cells were identified as CD19^+^/CD5^+^ cells. The frequency of blood CD19^+^/CD5^+^ cells was 80 ± 4% (mean ± SEM), and 68 ± 10% of the blood MCL cells exhibited surface ROR1.

### 3.2. KAN0441571C Induced Significant Cell Death of MCL Cell Lines

KAN0441571C induced a time- and dose-dependent cell death (MTT) of all MCL lines but to varying degrees. Seventy-two hours of incubation were required to achieve the optimal cytotoxic effect. JeKo1 seemed to be the most sensitive cell line and Z138 the least sensitive (Figure 1A). Next, apoptosis analysis (Annexin V/PI staining) was performed after incubation with KAN0441571C, venetoclax, or bendamustine (Figure 1B and Appendix A). JeKo1 was again the most sensitive cell line to ROR1 inhibition, reaching almost 100% killing at 250 nM, while venetoclax and bendamustine were less effective. A similar pattern was seen for Mino, Z138, and JVM-2, while Granta-519 was the most sensitive to venetoclax. The data support the notion that ROR1 inhibition by a small molecule inhibitor seems to effectively induce death of MCL cells. We analyzed protein phosphorylation and found, as expected, that KAN0441571C prevented phosphorylation of ROR1 in a dose-dependent manner in all MCL cell lines (Appendix A). Previous studies of ROR1 inhibitors in tumor cells from other hematological malignancies have indicated the involvement of both WNT non-canonical (PI3K/AKT/mTOR) and WNT canonical (beta-catenin) signaling pathways, as well as downregulation of the intrinsic apoptotic pathway, including cleavage of caspase 3 [12,27,28]. When primary MCL cells were analyzed, we also found that in addition to the prevention of ROR1 phosphorylation, PI3K, AKT, and mTOR were also subsequently dephosphorylated (Figure 2).

### 3.3. Effects of KAN0441571C in Combination with Other Drugs

First, we evaluated the apoptotic effect of single-drug KAN0441571C, venetoclax, ibrutinib, idelalisib, everolimus, or bendamustine on patient-derived blood MCL cells (n = 11). Dose–response curves for each individual drug, including EC_50_ values, are shown in Figure 3. Venetoclax and KAN0441571C were the most effective single agents among the tested drugs. Next, MCL cell lines were incubated with the EC_50_ concentration of KAN0441571C and equimolar concentrations of venetoclax, ibrutinib, or bendamustine, alone and in combination (Appendix A). In single-drug experiments, KAN0441571C, venetoclax, and ibrutinib were similarly effective in inducing cell death of the four different MCL cell lines, while bendamustine seemed to be inferior in two of five cell lines (Mino and Granta-519). In drug combination experiments, KAN0441571C plus venetoclax or KAN0441571C plus ibrutinib showed a statistically significant increased cytotoxic effect, indicating synergism for three cell lines (not for Granta-519) compared to each drug alone, while the combination of KAN0441571C with bendamustine showed synergism in all cell lines except for Granta-519.

We then repeated the drug combination experiments using patient-derived MCL cells. The results are shown in Figure 4. There was a significant increase in apoptosis (Annexin V/PI) of MCL cells when KAN0441571C was incubated in combination with either venetoclax, ibrutinib, idelalisib, everolimus, or bendamustine as compared to each drug alone. Furthermore, when drugs were combined, low doses of each drug seemed to be as effective as high doses.

A summary of apoptosis ranking for all drug combinations in primary MCL cells is displayed in Figure 5. The highest synergism was seen between a low dose (50 nM) of KAN0441571C and a low dose (5000 nM) of idelalisib. Moreover, a low dose (50 nM) of KAN0441571C and a low dose (5 nM) of venetoclax showed significant synergism. A high dose (100 nM) of KAN0441571C and everolimus (20,000 nM) induced a significant synergistic effect, as well. To confirm apoptosis, we analyzed the expression of BCL-2 and MCL-1, which were markedly downregulated by the combination of drugs as compared to each drug alone, while cleaved caspase 3 and cleaved PARP were upregulated (Figure 6 and Appendix A).

As shown above, KAN0441571C alone dephosphorylated ROR1, followed by the inactivation of AKT, BTK, and mTOR, which are involved in the ROR1 signaling pathway (Appendix A and Figure 2). When KAN0441571C was combined with venetoclax, ibrutinib, everolimus, or idelalisib, phosphorylation of ROR1, AKT, BTK, and mTOR was decreased as compared to single drugs (Figure 2). The most effective combination seemed to be KAN0441571C with venetoclax or ibrutinib. Interestingly, phospho-PI3K was least affected by all drugs and combinations including idelalisib (Figure 2). Overall, the pattern of phosphorylation of target proteins was less clear as compared to that of apoptotic proteins, probably reflecting the complex network of intracellular signaling.

## 4. Discussion

The introduction of new drugs has improved the prognosis for patients with MCL. However, there is an urgent medical need for novel treatment options to increase the response rate and depth of remission and a drug with other MOAs than those currently used at clinical relapse and resistance. ROR1 has been proposed to be a druggable target [26,31] and is reported to be heterogeneously expressed in MCL [33]. In this study, we confirmed and extended the expression pattern of ROR1 in MCL cell lines and primary MCL cells. There are also splice variants of ROR1. A truncated ROR1 lacking the extracellular domain with an intact intracellular region has been described [2]. Forouzesh et al. [34] also reported a splice variant which only had the extracellular region. These results might explain the findings in JeKo1 and Mino cell lines comparing surface staining and Western blot. Different isotypes of ROR1 have also been reported with molecular weights from 100 to 130 kDa, which may represent different degrees of glycosylation [6,34].

KAN0441571C is the second generation of a small molecule ROR1 tyrosine kinase inhibitor [30]. The drug induced apoptosis of ROR1^+^ MCL cell lines as well as primary MCL cells. The ROR1 inhibitor prevented phosphorylation of ROR1. Molecules of the non-canonical WNT signaling pathway were inactivated, which is in line with previous reports showing that ROR1 signals through at least the β-catenin-independent (non-canonical) WNT pathway [35,36,37]. Involvement of the PI3K/AKT/mTOR pathway in MCL is also supported by a previous report showing that inhibition of mTOR in MCL cells induced cell cycle arrest and apoptosis [38]. Furthermore, AKT activates mTOR through the interaction of several signaling molecules [35], including the downstream target CREB. AKT inactivation is of importance for apoptosis induced by many targeting drugs [39]. The ERK pathway, which is frequently mutated in malignancies [40], was also inactivated by ROR1 inhibition, which is in agreement with a report showing that ERK was dephosphorylated followed by treatment with KAN0441571C in small-cell lung cancer cells [41]. Moreover, ROR1 interacts with SRC, a key regulator of cancer cells. KAN0441571C prevented phosphorylation of SRC, which should also be of significance for induction of tumor cell death.

Intracellular signaling is a complex network of interactions between proteins. Malignant cells are masters to overcome signal suppression through upregulation of other signaling proteins. In MCL, various signaling pathways seemed to be used, which may vary between patients. A drug which can act on several key oncogenic signaling molecules should be of interest in a therapeutic attempt. KAN0441571C not only prevented phosphorylation of ROR1 but also inactivation of the PI3K/AKT/mTOR molecules, an important axis in tumorigenesis and a significant therapeutic target [42].

The combination of KAN0441571C with other agents had a synergistic apoptotic effect. Increased tumor cell death induced by KAN0441571C in combination with other drugs was supported by enhanced downregulation of, e.g., the anti-apoptotic MCL-1 and BCL-2 proteins, which are highly expressed in MCL cells [34]. The BCL-2 family proteins are crucial regulators of the mitochondrial apoptotic pathway. Genetic aberrations of these genes correlated with lymphomagenesis and chemotherapy resistance [33,43]. Altering the balance between anti-apoptotic and pro-apoptotic BCL-2 proteins may lead to the evasion of apoptosis and the extension of the survival of malignant cells. Targeting MCL-1 and BCL-2 might be a rewarding therapeutic approach in MCL.

Drugs which are in clinical use for MCL such as ibrutinib, venetoclax, idelalisib, everolimus, and bendamustine showed significant killing of MCL cells. The ROR1 inhibitor KAN0441571C seemed to act synergistically with these drugs, enhancing tumor cell death. Of specific interest was that low doses of KAN0441571C and low doses of venetoclax, ibrutinib, and idelalisib had a significant synergistic apoptotic effect. The increased apoptotic effect of KAN0441571C in combination with other targeting drugs might be due to enhanced inactivation of various signaling molecules associated with ROR1 signaling.

This is the first study on the effects of a small molecule ROR1 inhibitor alone on MCL tumor cell survival and signaling. The data also indicated that combining the ROR1 inhibitor with drugs targeting other dysregulated pathways contributes to an increased anti-tumor effect. Previous studies in diffuse large B-cell lymphoma (DLBCL) and small-cell lung cancer (SCLC) have indicated that combining ROR1 inhibitors with venetoclax [30,41,44] or with erlotinib and chemotherapeutics in pancreatic carcinoma [5] acted synergistically to induce tumor cell death. This report also adds PI3K and BTK inhibitors to the group of combinatorial partners for ROR1 inhibitors. Further pre-clinical and clinical studies are warranted to evaluate this new treatment concept with respect to side effects and anti-tumor activity. As the tumor microenvironment is of importance for tumor progression [45], models where a tumor microenvironment is incorporated should be included. There is a substantial amount of data indicating the importance of the microenvironment in MCL [26]. We have previously shown that when CLL cells were cultured in the presence of stromal cells, low concentrations of the ROR1 inhibitor were not as effective in killing leukemic cells as if cultured without stromal cells. When a high concentration of the inhibitor was used, stromal cells had no preventive effect [29].

## 5. Conclusions

ROR1 is an oncogenic RTK involved in the survival of tumor cells of various origins. Due to its unique expression in tumors, ROR1 has been recognized as an interesting target for cancer treatment. ROR1 small molecule inhibitor in hematologic malignancies, including MCL, induced significant apoptosis of tumor cells in vitro as well as the inhibition of several important signaling pathways for tumor cell survival. In combination with other small molecules targeting MCL tumor cells, a significantly increased tumor cell death was seen.

## Figures and Tables

**Figure 1 pharmaceutics-14-02238-f001:**
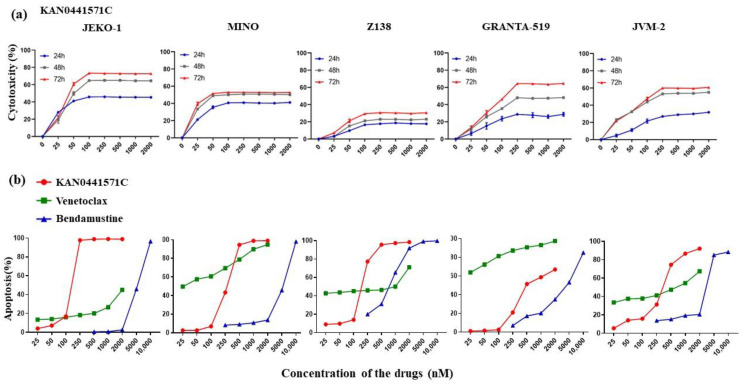
**Effects of KAN0441571C, venetoclax, and bendamustine on MCL cell lines.** (**a**) Cytotoxicity (MTT) of 5 ROR1^+^ MCL cell lines incubated with the ROR1 inhibitor KAN0441571C for various time periods. (**b**) Apoptosis (Annexin V/PI) of MCL cell lines incubated with KAN0441571C, venetoclax, or bendamustine alone for 48 h. Data are shown as mean ± SEM of triplicate samples.

**Figure 2 pharmaceutics-14-02238-f002:**
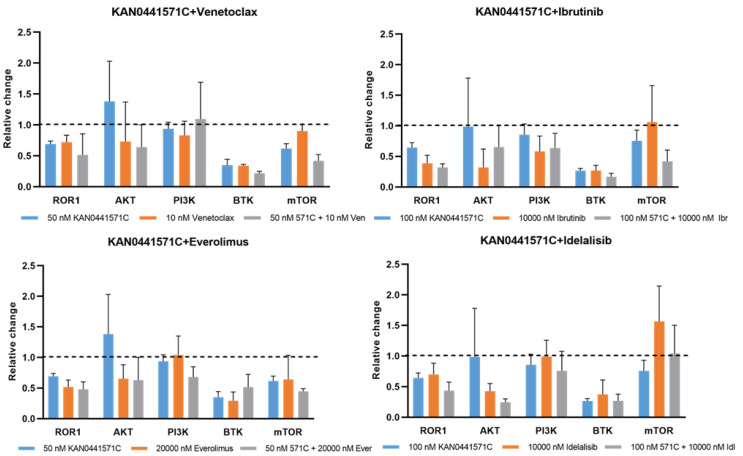
**Effects of KAN0441571C, venetoclax, ibrutinib, and idelalisib as well as the combinations on intracellular signaling molecules.** Relative change in primary MCL cells in the levels of phosphorylated ROR1, AKT, PI3K, BTK, and mTOR after single-drug incubation with KAN0441571C (571C) and venetoclax (Ven), ibrutinib (Ibr), everolimus (Ever), and idelalisib (Idel), respectively, and in combination, as indicated at the bottom of the figure. The dotted lines indicate the level of the phosphorylated protein in untreated cells. The figure shows mean ± SEM of 3 patients.

**Figure 3 pharmaceutics-14-02238-f003:**
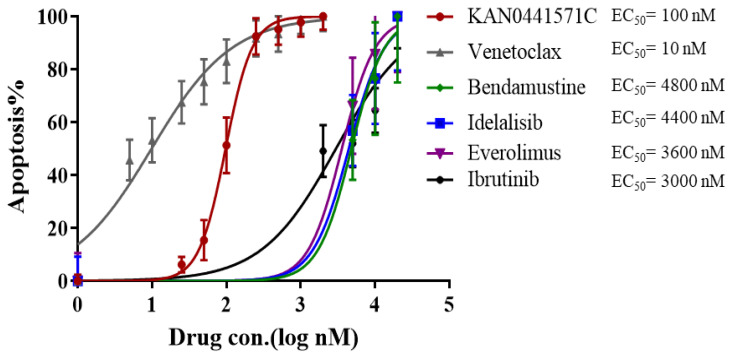
**Dose–response curves and EC_50_ values for each individual drug.** Apoptosis (Annexin V/PI) (mean ± SEM) of MCL cells from patients (n = 11) incubated in vitro with KAN0441571C, venetoclax, bendamustine, idelalisib, everolimus, and ibrutinib for 24 h. EC_50_ values for each drug are shown. R-square values are 0.119, 0.6161, 0.2826, 0.3542, 0.3077, and 0.5438 for KAN0441571C, venetoclax, bendamustine, idelalisib, everolimus, and ibrutinib, respectively.

**Figure 4 pharmaceutics-14-02238-f004:**
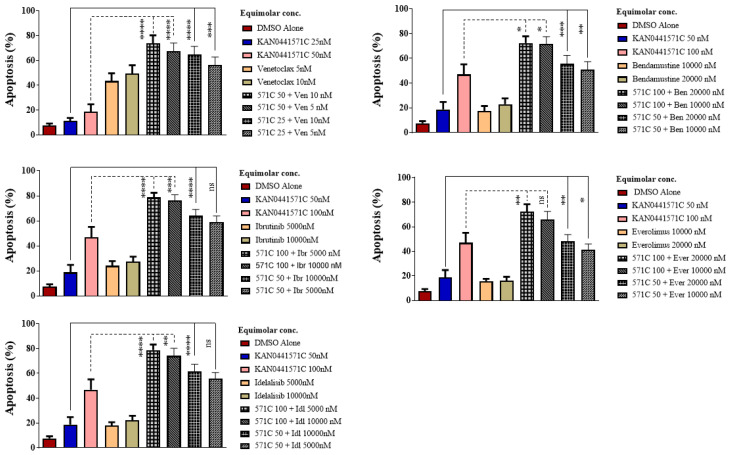
**Induction of apoptosis by inhibitors alone or in combination on primary MCL cells.** Apoptosis (Annexin V/PI) (mean ± SEM) of primary MCL cells (n = 11) incubated in vitro with equimolar concentrations of KAN0441571C (571C), ibrutinib (Ibr), venetoclax (Ven), idelalisib (Idl), everolimus (Ever), or bendamustine (Ben) alone and in combination. Statistical significances are shown at the top: * *p* < 0.05, ** *p* < 0.01, *** *p* < 0.001, **** *p* < 0.0001; ns: not significant (Mann–Whitney test was used for comparisons between different groups).

**Figure 5 pharmaceutics-14-02238-f005:**
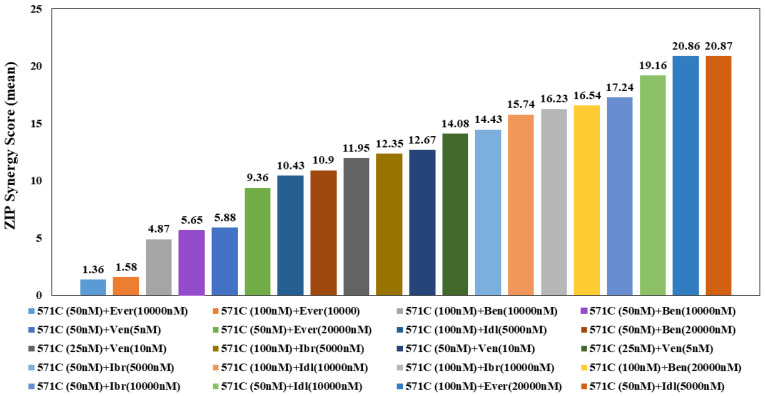
**Ranking (ZIP synergy score) for apoptosis induced by drug combinations in primary MCL cells.** ZIP synergy score (mean) for apoptosis (Annexin V/PI) of drug combinations using MCL cells derived from patients (n = 11) incubated for 24 h by indicated drugs and conc. The ZIP score captures the effect of drug combinations by comparing the change in the potency of the dose–response curves between individual drugs and their combination (https://synergyfinder.org/ (accessed on 21 August 2021)). A score between 1 and 10 indicates an additive effect and >10 synergism. KAN0441571C (571C), everolimus (Ever), ibrutinib (Ibr), idelalisib (Idl), bendamustine (Bend), venetoclax (Ven).

**Figure 6 pharmaceutics-14-02238-f006:**
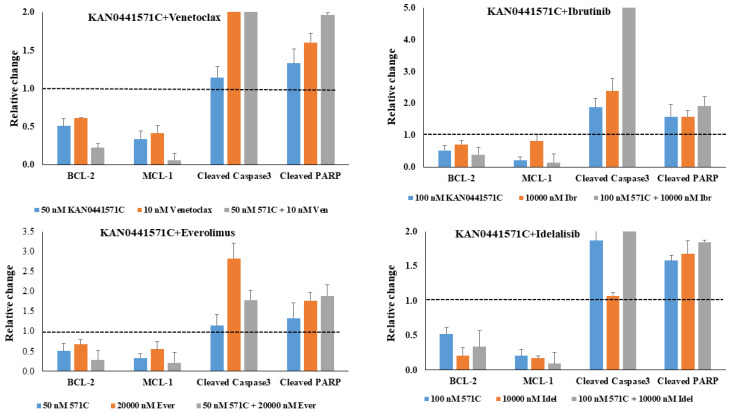
Induction of apoptosis following treatment of MCL primary cells with KAN0441571C, venetoclax, ibrutinib, everolimus, or idelalisib. Relative change in MCL cells from patients of BCL-2, MCL-1, cleaved caspase 3, and cleaved PARP after incubation with single-agent KAN0441571C (571C), venetoclax (Ven), ibrutinib (Ibr), everolimus (Ever), or idelalisib (Idel) alone and in combination, as indicated at the bottom of the figure. The dotted lines indicate the level of untreated cells. The figure shows the results of one representative patient out of three.

## Data Availability

Not applicable.

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
