# Peer review of "A ROR1 Small Molecule Inhibitor (KAN0441571C) Induced Significant Apoptosis of Mantle Cell Lymphoma (MCL) Cells"

_pharmaceutics, 2022, doi:10.3390/pharmaceutics14102238_

Round 1

Reviewer 1 Report (Previous Reviewer 2)

Amineh Ghader uncovered an exciting topic regarding MCL. Points to be addressed: 

1. I would suggest to slightly restructure the manuscript as follows:

P (Population or problem)

Who or what is the patient, population or problem in question?

I (Intervention)

What is the intervention (action or treatment) being considered?

C (control)

What other interventions should be considered?

O (Outcome or objective)

What is the desired or expected outcome or objective?

T (treatment)

What is the treatment and time of observation of their experimental setting?

2. Did the authors check for normal (gaussian) distribution? If this is the case this should be clearly explained, otherwise the authors should employ non parametric tests.

3. The rationale of why the authors came up with this experimental settings: microenvironment is totally neglected in the introduction section, please expand. 

4.What is the information that is not exactly available that motivated the authors to come up with this information. What are the current caveats and how do the authors highlight the current research in answering them? If not they need to address in future directions.

5.The authors need to highlight what new information the review is providing to enhance the research in progress (please summarize with a graphical abstract the workflow, or (alternatively, on top of the figures briefly explain graphically the experimental setting)

6. The authors could provide a little more consideration of genomic directed stratifications in clinical trial design and enrollments.  

7. The underlying message here is that more precision and individualized approaches need to be tested in well designed clinical trials – a challenge, but I would be interested in their perspective of how this might be done.

8. mantle cell lymphoma and other lymphoma subtypes often spread to the bone marrow, and stromal interactions mediated by focal adhesion kinase frequently enhance survival and drug resistance of the lymphoma cells: please refer to PMID: 29079592 and expand the introduction/discussion sections.

Author Response

Reviewer 1

Amineh Ghader uncovered an exciting topic regarding MCL. Points to be addressed

Many thanks for your complement.

  1. I would suggest to slightly restructure the manuscript as follows:

Response: A new structure of the manuscript as suggested by the reviewer is an interesting idea. We however feel that it may fit better in the context of a clinically oriented review article, or possibly for a manuscript containing clinical data. As the current manuscript is entirely dealing with pre-clinical drug experiments, we suggest keeping its traditional layout. we hope that would be acceptable for the reviewer. 

P (Population or problem)

Response: See above regarding structure of the manuscript. Additional clarification regarding the unmet need among patients with MCL has been made in the Introduction (page 2, lines 67-92) and the Discussion (page 10, 11, lines 293-295 and 353-365).

Who or what is the patient, population or problem in question?

What is the intervention (action or treatment) being considered?

I (Intervention)

Response: See above regarding the structure of the manuscript. This is a pre-clinical study meaning that clinical intervention is a later step.

C (control)

Response: As this is a preclinical drug evaluation, appropriate control experiments are reported in the Methods and Results sections.  Selection of clinical control groups is a much later question when intervention trials will be initiated.

What other interventions should be considered?

O (Outcome or objective)

Response: See above. This preclinical manuscript uses laboratory outcome analysis as reported in the Methods and Results sections. Outcome/objective of clinical intervention trials is the subject of a later study.

What is the desired or expected outcome or objective?

T (treatment)

What is the treatment and time of observation of their experimental setting?
Response: See above. Clinical intervention is a much later step.

  1. Did the authors check for normal (gaussian) distribution? If this is the case this should be clearly explained, otherwise the authors should employ non-parametric tests.

Response: We tested the distribution of our data and as the distributions were not normal, we used man-Whitney test (non-parametric) for comparisons between different groups. This statement is added to the statistical analysis (page 4, line 181).

  1. The rationale of why the authors came up with these experimental settings: microenvironment is totally neglected in the introduction section, please expand.

Response:  The experimental settings represent standard methods in this type of early pre-clinical testing effects of cytotoxic drugs. We agree with the reviewer that interference with the tumor microenvironment (TME) is another emerging therapeutic possibility. A comment has been added in the Introduction (page 2, lines 69-72).

4.What is the information that is not exactly available that motivated the authors to come up with this information. What are the current caveats and how do the authors highlight the current research in answering them? If not, they need to address in future directions.

Response:  MCL is a ROR1 expressing malignancy with unmet clinical needs and suggested to be included in the evaluation of ROR1 targeting drugs.  This is the first preclinical evaluation of a ROR1-targeting small molecule with considerable novel information supporting the notion to continue its preclinical and hopefully later also clinical development in MCL.

5.The authors need to highlight what new information the review is providing to enhance the research in progress (please summarize with a graphical abstract the workflow, or (alternatively, on top of the figures briefly explain graphically the experimental setting).

Response: A graphical abstract has been added to better describe how the new information has been addressed in this research paper.

  1. The authors could provide a little more consideration of genomic directed stratifications in clinical trial design and enrolments.  

Response:  This kind of information is not the focus of the present preclinical manuscript but should be considered when planning a clinical study to identify subsets of patients who may likely respond or not respond to the drug when/if a drug candidate may reach clinical phase 2 and 3 studies. We prefer not to speculate on impact of genomic profiling at this early preclinical drug development stage.

  1. The underlying message here is that more precision and individualized approaches need to be tested in well-designed clinical trials – a challenge, but I would be interested in their perspective of how this might be done.

Response:  We agree with the reviewer, but this question is raised too early. It this stage, the ROR1 small molecule therapeutic approach is at an early pre-clinical development stage. Thus, design of clinical trials is a later step but should, of course, relate clinical efficacy to baseline factors, including but not limited to ROR1 intensity expression. We suggest to omit such a discussion in the current early preclinical paper.

  1. mantle cell lymphoma and other lymphoma subtypes often spread to the bone marrow, and stromal interactions mediated by focal adhesion kinase frequently enhance survival and drug resistance of the lymphoma cells: please refer to PMID: 29079592 and expand the introduction/discussion sections.

Response: The reference has been added and the Introduction section has been expanded (Reference 22, page 2, lines 69-72).

Reviewer 2 Report (New Reviewer)

The first ROR1 small molecule inhibitor was developed 4 years ago. Accordingly, the development of ROR1 small molecule inhibitors could attract much attention from researchers in the near future. In the current study, KAN0441571C was evaluated in vitro and ex vivo preclinical models alone or in combination with other drugs, which act with different mechanisms of action. However, I have a few comments below:

·         The manuscript (PDF) file I have received includes track changes colored in yellow and brown (Lines: 90, 91, 171, 182, 184, ….. etc). I  think the authors or the editorial office may make these corrections.

·         The authors published a previous study in 2019, discussing the effect of KAN0441571C alone or in combination with the Bcl-2 inhibitor venetoclax, Blood, 134, p. 5312. The study revealed that “The combination of ROR1 inhibitor and venetoclax had a synergistic apoptotic effect. Why did the authors not cite this work in the current study?

·         The reasons beyond the selection of venetoclax, bendamustine, idelalisib, ….. for the combination with KAN0441571C could be helpful for readers if it was discussed at the beginning of the manuscript.

·        A molecular docking study to identify the potential binding site/mode of KAN0441571C could enrich the current study. 

·         line 65: “Hematological malignancies are the seventh most common type …… ” a reference should be added to this sentence.

·         Line 198: Figure 1: 5 cell lines. Is it 4 or 5 cell lines?

·         The treatment time in the Annexin V/PI apoptosis assay was 24 hr, while in the MTT assay it was 24-72 h.

Author Response

Reviewer 2 comments

The first ROR1 small molecule inhibitor was developed 4 years ago. Accordingly, the development of ROR1 small molecule inhibitors could attract much attention from researchers in the near future. In the current study, KAN0441571C was evaluated in vitro and ex vivo preclinical models alone or in combination with other drugs, which act with different mechanisms of action. However, I have a few comments below:

  • The manuscript (PDF) file I have received includes track changes coloured in yellow and brown (Lines: 90, 91, 171, 182, 184, …. etc). I think the authors, or the editorial office may make these corrections.

Response: We had corrected the manuscript due to the comments of 3 earlier reviewers, as suggested when submitted to Pharmaceutics.  Track changes related to the 1st series of comments have been omitted in the new resubmitted version.

  • The authors published a previous study in 2019, discussing the effect of KAN0441571C alone or in combination with the Bcl-2 inhibitor venetoclax, Blood, 134, p. 5312. The study revealed that “The combination of ROR1 inhibitor and venetoclax had a synergistic apoptotic effect. Why did the authors not cite this work in the current study?

Response: Thank you for this comment. We have added the reference to the manuscript.

  • The reasons beyond the selection of venetoclax, bendamustine, idelalisib, ….. for the combination with KAN0441571C could be helpful for readers if it was discussed at the beginning of the manuscript.

Response: Clarification on selection of drugs had been added in the Introduction (page 2, lines 87-90).

  •       A molecular docking study to identify the potential binding site/mode of KAN0441571C could enrich the current study. 

Response: In structural alignment, the docked confirmation of the first class of our ROR1 inhibitor (KAN0439834) showed that it matched well with the ATP-binding pocket of ROR1 of the TK domain with the central scaffold of KAN0439834 forming h-bond in the hinge region. KAN0441571C has a similar structure. Tese information is added in reference 31.

  • line 65: “Haematological malignancies are the seventh most common type …… ” a reference should be added to this sentence.

Response: The statement is corrected, and a reference has been added.

  • Line 198: Figure 1: 5 cell lines. Is it 4 or 5 cell lines?

Response: We corrected the statement. 5 cell lines is correct.

  • The treatment time in the Annexin V/PI apoptosis assay was 24 hr, while in the MTT assay it was 24-72 h.

Response: As apoptosis occurs prior to cell death, we have to run cytotoxicity assay after 48-72 hours with timepoints selected based on the biological nature of each cell line.

Round 2

Reviewer 1 Report (Previous Reviewer 2)

I am satisfied with the rebuttal provided.

This manuscript is a resubmission of an earlier submission. The following is a list of the peer review reports and author responses from that submission.

Round 1

Reviewer 1 Report

The aim of this study is to identify the role of “the receptor tyrosine kinase-like orphan receptor 1 (ROR1) small molecule inhibitor (KAN0441571C) induced significant apoptosis of mantle cell lymphoma (MCL) cells”. The concept is interesting, however, the authors failed to do the experiment correctly.

Comments:

  1. The abstract is weak, and the research goal is missing.
  2. Authors should use headlines in all figure legends.
  3. In figure 1: Each concentration does not have a triplicate. There are no error bars and statistical analyses. Before attempting an experiment, authors must plan carefully. Please see how other writers presented their figures in a similar study from a cancer journal or another journal. Please repeat the experiment and do the statistical analysis.
  4. In figure 2: Same comments as figure 1.
  5. In figure 3. Please do statistical analysis.
  6. In figure 4: The authors claim statistically significant but in the figure, there are no P-value and star symbols (*, **, ***).
  7. In figure 5: Same comments as figure 1.
  8. In figure 6: Same comments as figure 1.
  9. S figure S1: Authors should use different concentrations of KAN0441571C. Please repeat the experiment using positive and negative control.
  10. S figure S2: Authors should add western blot band quantification. It is difficult to distinguish differences without band quantification.
  11. S figure S4: How much protein is used per loading and what is the MW of the proteins. Please add band quantification result.
  12. S figure S5: Western bot experiment is not conducted correctly in case of mTOR and pmTOR. The band must be in the same blot, but the authors used another cutting band. It looks serious flaws in the experiment. Please never do that.

Reviewer 2 Report

Amineh Ghaderi et al. uncover ROR1 as a small molecule inhibitor (KAN0441571C) to be potentially able to induce significant apoptosis of mantle cell lymphoma (MCL) cells.

Point to be considered:

  1. Did the author employ unstained/isotype controls for flow cytometry? Plots should be representatively showed.
  2. The authors explained Western blot (WB) analysises: for all Western blot figures, densitometry readings/intensity ratio of each band should be included; the whole Western blot showing all bands and molecular weight markers should be included in the Supplementary Materials; 

    3. This reviewer personally misses some insights regarding mantle cell lymphoma and other lymphoma subtypes propensity to spread to the bone marrow, and stromal interactions mediated by focal adhesion kinase frequently enhance survival and drug resistance of the lymphoma cells: focal adhesion kinase is highly expressed in bone marrow infiltrates of mantle cell lymphoma and in mantle cell lymphoma cell lines. Stroma-mediated activation of focal adhesion kinase led to activation of multiple kinases (AKT, p42/44 and NF-κB), that are important for prosurvival and proliferation signaling (please refer to PMID: 29079592 and expand according to the relevance for the authors findings.)

Reviewer 3 Report

See attachment for comments.

Round 2

Reviewer 1 Report

Dear Authors,

Thank you for the revision.

I double-checked the manuscript and entire blots file, however, there are still cutting blots and gapdh/beta-actin missing in each blot. You weren't able to present entire blots with ladder, and each blot should have gapdh/beta-actin.

How did you know your experiment was successful without a housekeeping gene in all blot?

It is very hard to accept your data.

Reviewer 3 Report

I am okay with the answers provided for comments 1, 2.1, 2.2, 2.4, 3.3, 3.75 and 3.8.1.

However, I totally don't agree on the rebuttal provided for comment 3.1. As can be seen from figure 1a, even after 72 h exposure, the maximal kill/inhibition of proliferation of KAN against Z138 is <100%. This makes your calculation of EC50 = 500 nM, unacceptable, and necessitates the redo of the experiment. The other answer you provided is also totally misleading and unacceptable for a reviewer with cancer cell biology expertise. Using a cell line with varying passage number will never produce such variation on the sensitivity of a cell line. Definitely, it won't change an active compound (as in the case in apoptosis) into inactive compound (as in the case in MTT result) for Z138. In conclusion, the EC50 for KAN against Z138 can't be <2000 nM from the dose-response curve you provided (Fig 1a). Moreover, on the same cell line, your compound killed only <40% of cells, how can it cause a near 100% kill at 500 nM (fig 1b). This shows that at least one of the results is unacceptable and the experiment shall be redone. 

All the original flowcytometry graphs should be presented for those figures presented as bar graphs (fig 3, 4 and 5) as supplementary data. The data for the negative control (cells treated with 10 uL DMSO) shall be included in each graph rather than subtracted. It is not a blank data, and shall not be subtracted from the treatment. The data from the negative control not only is used to compare results with the treatment but also can tell if the sample processing was properly done. Moreover, 10 ml DMSO as a control seems too much. Example, in your MTT assay you added 200uL media, and 10 mL DMSO in this media will give 5% DMSO concentration, which is too high for some cell lines and can cause cytotoxicity by itself. In general, the final DMSO concentration is not recommended to exceed 1%.

For 3.4, the original annexin V/PI data provided and the value presented on Fig 3 don't match. For 3.5, the values for the negative control should be presented, and the type of statistics used should be stated in every legend where statistical comparison is used.